# Auto-Compressing Networks

**Vaggelis Dorovatas**[1,2]
vdorovatas@hotmail.gr

**Georgios Paraskevopoulos**[3]
g.paraskevopoulos@athenarc.gr

**Alexandros Potamianos**[1,2]
potam@central.ntua.gr

[1]National Technical University of Athens
[2]Archimedes RU, Athena RC
[3]Institute of Language and Speech Processing, Athena RC

## Abstract

Deep neural networks with short residual connections have demonstrated remarkable success across domains, but increasing depth often introduces computational redundancy without corresponding improvements in representation quality. We introduce Auto-Compressing Networks (ACNs), an architectural variant where additive long feedforward connections from each layer to the output replace traditional short residual connections. By analyzing the distinct dynamics induced by this modification, we reveal a unique property we coin as *auto-compression*—the ability of a network to organically compress information during training with gradient descent, through architectural design alone. Through auto-compression, information is dynamically "pushed" into early layers during training, enhancing their representational quality and revealing potential redundancy in deeper ones. We theoretically show that this property emerges from layer-wise training patterns present in ACNs, where layers are dynamically utilized during training based on task requirements. We also find that ACNs exhibit enhanced noise robustness compared to residual networks, superior performance in low-data settings, improved transfer learning capabilities, and mitigate catastrophic forgetting suggesting that they learn representations that generalize better despite using fewer parameters. Our results demonstrate up to 18% reduction in catastrophic forgetting and 30-80% architectural compression while maintaining accuracy across vision transformers, MLP-mixers, and BERT architectures. These findings establish ACNs as a practical approach to developing efficient neural architectures that automatically adapt their computational footprint to task complexity, while learning robust representations suitable for noisy real-world tasks and continual learning scenarios.

## 1 Introduction

Deep learning has achieved significant breakthroughs across diverse tasks and domains [28, 30, 6]; however, it still lacks the flexibility, robustness, and efficiency of biological networks. Modern models rely on deep architectures with billions of parameters, leading to high computational, storage, and energy costs. Architecturally, these large models are primarily characterized by short residual connections [20], a design initially developed to enable robust training of deep neural networks via backpropagation by providing stable gradient flow through these exceptionally deep networks. These skip connections establish a network topology where multiple information pathways are created [53], resulting in an ensemble-like behavior that delivers more efficient training and superior generalization compared to traditional feedforward networks.

39th Conference on Neural Information Processing Systems (NeurIPS 2025).

Historically, since the emergence of Highway Networks [47], which first proposed additive skip connections researchers have explored numerous architectural variations. Residual Networks (ResNets) [20] removed learned gating functions and adopted direct identity skip connections becoming the industry standard. DenseNets [22] utilized feature concatenation instead of addition, while FractalNets [29] introduced a recursive tree-like architecture combining subnetworks of multiple depths to further enrich feature fusion. More recent works include learned weighted averaging across layer outputs [37], application of attention mechanisms across block outputs [14] and denser connectivity patterns between network nodes [63]. Other works have explored adding scalars to either the residual or block stream to improve performance, training stability and representation learning [43, 3, 62, 16]. In neural machine translation, researchers have drawn inspiration from both vision and language domains to combine information from different layers, enabling richer semantic and spatial propagation throughout the network [11, 59].

| Arch | Connectivity | Forward Propagation | Backward (Gradient) Propagation |
|---|---|---|---|
| FFN | | $y_F = \prod_{i=1}^{L} w_i x_0$ | $\frac{\partial y_F}{\partial w_i} = \underbrace{\left( \prod_{k=i+1}^{L} w_k \right)}_{backward\ term} \underbrace{\left( \prod_{m=1}^{i-1} w_m \right)}_{forward\ term} x_0$ |
| ResNet | | $y_R = \prod_{i=1}^{L} (1 + w_i) x_0$ | $\frac{\partial y_R}{\partial w_i} = \underbrace{\left( \prod_{k=i+1}^{L} (1 + w_k) \right)}_{backward\ term} \underbrace{\left( \prod_{m=1}^{i-1} (1 + w_m) \right)}_{forward\ term} x_0$ |
| ACN | | $y_A = \left( 1 + \sum_{i=1}^{L} \prod_{j=1}^{i} w_j \right) x_0$ | $\frac{\partial y_A}{\partial w_i} = \underbrace{\left( 1 + \sum_{j=i+1}^{L} \prod_{k=i+1}^{j} w_k \right)}_{backward\ term} \underbrace{\left( \prod_{m=1}^{i-1} w_m \right)}_{forward\ term} x_0$ |

Table 1: Connectivity (2D case), Forward and Backward Propagation (1D linear case) for FFN, ResNet, and ACN architectures.

While Residual Networks, as discussed, offer efficient and effective training especially in very deep architectures, there are various works that explore behaviors of these architectures that may harm generalization. In [53, 48] it is shown that these architectures exhibit a notable resilience to layer dropping and permutation, which could indicate potential redundancy of some layers (i.e. removing a layer does not affect the network). In [23], it was further observed that dropping subsets of layers during training can reduce overfitting and improve generalization. In a related study [1], the authors showed that introducing skip connections between layers can lead to parts of the network being effectively bypassed and under-trained. In [62], they show that unscaled residual connections can harm generalization capabilities in generative representation learning. Finally, more recently, research has revealed substantial parameter redundancy and inefficient parameter usage in large-scale foundation models, particularly within their deeper layers [18, 7]. All these observations can be unified under the perspective that, although residual architectures facilitate training via multiple signal pathways, these same pathways can sometimes act as shortcuts that cause certain components to be either underutilized or prone to overfitting—ultimately limiting effective generalization.

Inspired by these observations, in this work we explore whether we can design alternative architectures that preserve the key advantages of Residual Networks—such as multiple signal pathways and efficient gradient flow—while addressing issues like redundancy and shortcut overuse, ultimately enabling more efficient parameter utilization and improved representation learning. To this end, we propose an architectural variant where additive long feedforward connections from each layer to the output replace traditional short residual connections as shown in Table 1, introducing Auto-Compressing Networks (ACNs). ACNs showcase a unique property we coin as *auto-compression*—the ability

of a network to organically compress information during training with gradient descent, through architectural design alone, dynamically pushing information to bottom layers, enhancing their representational quality, and naturally revealing redundant in deeper layers. We theoretically investigate the emergence of this property by analyzing the gradient dynamics of networks with different connectivity patterns. As illustrated in Figure 1, ACNs demonstrate layer-wise training patterns in which early layers receive significantly stronger gradients during the initial stages of training, in contrast to the more uniform gradient distribution observed in Residual Networks. Next, we empirically demonstrate a broad range of advantages that ACN-learned representations offer compared to residual or feedforward architectures, including: enhanced information compression, superior generalization, reduced catastrophic forgetting, and efficient transferability. Our contributions can be summarized as:

- We introduce Auto-Compressing Networks (ACNs), an architecture that organically compresses information into a subset of the full network's layers, through architectural design alone.

- We provide a detailed analysis of the gradient dynamics of ACNs, along with residual and feedforward networks, shedding light on their distinct behaviors and arguing that different connectivity patterns result in unique training regimes that drive distinct learned representations.

- We implement ACNs in fully connected and transformer-based architectures, finding that they match or outperform residual baselines, while 30–80% of top layers effectively become redundant as information concentrates in the lower layers. Notably, ACNs are hardware-friendly and require no specialized software.

- We show that ACNs learn representations that are more robust against noise and generalize better in low-data regimes compared to residual architectures.

- ACNs reduce catastrophic forgetting by up to 18% compared to residual networks in continual learning by preserving capacity for new unseen tasks.

- ACNs outperform regularization-based approaches at transfer learning without requiring hyperparameter tuning.

## 2   Auto-Compressing Networks

In ACNs [1], residual short connections are replaced with long feedforward connections, as described in Eq. 1 for a network of depth $L$, and shown [2] in Table 1:

$$x_i = f_i(x_{i-1}), \qquad y = \sum_{i=0}^{L} x_i \tag{1}$$

In ACNs the output of each layer is directly connected to the output of the network, and thus is directly optimized by the objective function during gradient descent training. Furthermore, the number of possible shortcuts is equal to the number of layers, $L$. We find this simplification maintains the improved signal flow that shortcut connections provide, while also introducing the ability to detect potential parameter redundancy in the architecture [3]. We note that ACNs differ structurally from other models employing long connections, such as DenseNets [22] and DenseFormer [37], which are residual networks variants, as previously discussed. These models connect each layer to all preceding layers whereas ACNs connect each layer only to the output, leading to a distinct structural design. An overview of these models along with an extended literature review is provided in Appendix A. For completeness, we also provide a direct, fair comparison against other residual variants, such as DenseFormers and DenseNets, in Appendix E.

---

[1]Code for the paper is available here.

[2]The connections shown for ResNet (red) are implicit, resulting from the residual summation. A detailed example is provided in Appendix D.

[3]Note that long connections are a strict subset of the $2^L$ shortcut connections in residual networks.

## 2.1 Gradient Propagation Across Network Architectures

In this section, we examine and compare the forward and backward pass (gradient flow) dynamics of three architectures: traditional feedforward networks (FFN), residual networks (ResNet), and the proposed auto-compressing networks (ACN), based on the equations of Table 1. See Appendix C for a detailed derivation of the gradient equations for 1D linear neural networks.

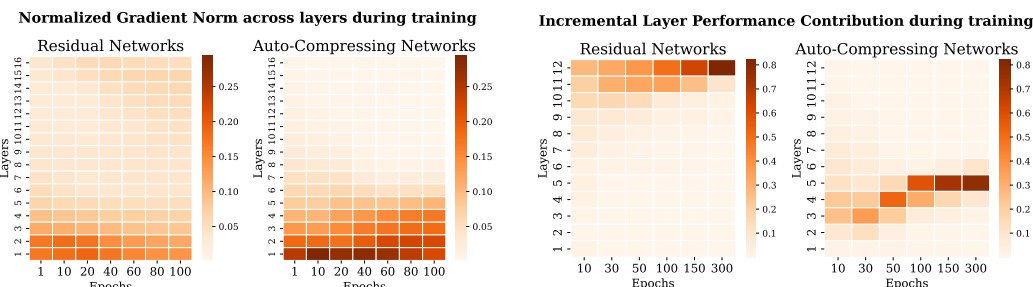

Figure 1: **(left)** ACNs vs Residual Networks gradient flow across layers during training for MLP-Mixer architecture [50] on CIFAR-10 [27], showcasing implicit layer-wise training and information concentration on the bottom layers for ACNs. **(right)** ACNs vs Residual Networks incremental performance contribution across layers for ViT architecture [10] on ImageNet-1K, revealing auto-compression by gradual layer-wise training in ACNs.

## 2.2 Emergent Gradient Paths

**The *forward* and *backward* components:** As shown in Table 1, each gradient $w_i$ (see $\frac{\partial y_*}{\partial w_i}$ in the 3rd column of the respective table) decomposes into forward and backward terms. The forward term determines gradient and forward propagation stability (whether the signal vanishes or explodes), while the backward term influences learning. For backward paths, 1D FFNs contain a single path, while 1D ACNs have $L - i + 1$: one direct path using the layer's own long connection to the output, plus $L - i$ additional paths where gradient flows from each subsequent layer's long connection and back through the network. 1D ResNets have $2^{L-i}$ paths since at each layer there are two options: flow through the network or follow the residual connection. It is also worth observing that ACNs feature a forward term identical to FFNs for intermediate layers (single path), while their backward components is closer to ResNets since it consists of multiple paths.

The Backward component (Full Gradient - **FG**) can be further decomposed into a *Network-mediated Gradient* (**NG**) component that is scaled by network weights and backpropagates information through (a subset of) the network and a *Direct Gradient* (**DG**) component that directly connects from the output to each layer, shown as the term "1" in the backpropagation equations of ACNs and ResNets in Table 1[4] This direct path acts as an information super highway, especially early in training where weights are typically initialized close to zero, informing each layer directly how to contribute towards lowering the optimization objective. Finally, the **DG** contribution is more significant for ACNs compared to ResNets, due to ACNs' linear (rather than exponential) total gradient path count.

Unlike the symmetric forward and backward terms of ResNets and FFNs, ACNs gradients, as argued, consist of a single forward path and multiple backward paths. This design creates an implicit layer-wise training dynamic, where deeper layers are trained at a slower rate compared to earlier layers, since they have a **weaker forward component** (assuming close-to-zero initialization) and a **smaller number of backward paths**. Further, when compared to ResNets, ACNs have a stronger contribution during backpropagations from the **DG** path (vs. **NG**) and this effect becomes more pronounced for deeper networks and for the early layers. For example, when training the second layer of a 1D $L = 12$ layer network, **DG** is one of 11 ACN backward paths, while for ResNets the **DG** is competing with another 127 paths (of the **NG** term). This further accelerates training of the early layers.

Thus, we postulate that: 1) a strong **DG** component coupled with a weaker feed-forward signal leads implicitly to efficient layer-wise training, and 2) architecturally-induced layer-wise training results

---

[4]This backward path has been previously explored as an alternative to traditional backpropagation and is typically referred to as Direct Feedback Alignment (DFA) in the literature [35, 40].

inadvertently in a form of **structural learning** where information is naturally pushed to early layers, i.e., later layers will become redundant (effectively identity mappings) if the earlier layers can already solve for the task. We refer to this new class of networks as **auto-compressors** since they naturally "shed" their redundant layers during backpropagation simply via architectural design. These claims are experimentally validated in the rest of the paper.

# 3 ACNs in Practice: Information Compression and Gradient Flow

Next, we implement auto-compressing networks on top of state-of-the-art neural architectures across diverse tasks and datasets. We implement ACNs using variants of the Transformer [52] for language and vision tasks and MLP-Mixer [50] for vision tasks. This allows us to evaluate our approach on diverse benchmarks including image classification (CIFAR-10 [27], ImageNet-1K [41]), sentiment analysis, and language understanding (BERT [9] on GLUE [54]).

In ACNs, for each input token, the final output vector $y^t$ is the sum of all intermediate layer representations plus the input embedding (Eq. 1). For classification, we apply pooling for images or use the `[CLS]` token for text. For a network of depth $L$, predictions using $k$ intermediate layers compute $y_k^t$ (sum of representations up to layer $k$) for each token, passed to a single shared classification head. This yields $L+1$ sub-networks, from input-only to full-network. For residual network baselines, at depth $k$, we take the output $y_k^t$ of layer $k$ as our $k + 1$ subnetwork's output.

Our experiments begin by empirically validating the main claim established in the previous section, i.e., the presence of a strong **DG** component coupled with implicit layer-wise training dynamics drives auto-compression, a property that resembles a form of structural (layer-wise) learning.

**Direct Gradient (DG) to total gradient ratio during training**

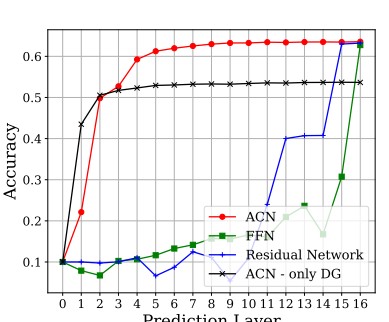
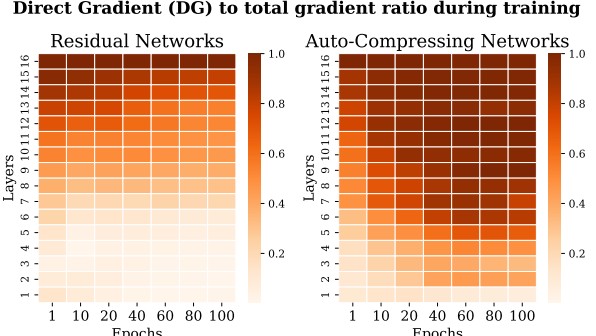

Figure 2: Results for the CIFAR-10 task using the MLM-Mixer base architecture: **(left)** ACN variants achieve auto-compression needing only a few layers to achieve good performance. **(right)** The ratio of direct gradient **DG** to the total gradient **FG** is significantly higher in early layers for ACNs.

To this end, We train feedfoward (FFN), residual and auto-compressing variants incorporated in the MLP-Mixer architecture on CIFAR-10 dataset for 100 epochs. To emphasize the role of the DG gradient in auto-compression, we also train an ACN variant receiving gradients only from the long connections (ACN - only DG component). In Figure 2(left), we show classification accuracy plotted against network depth (layer probing) and observe that among ACNs, FFNs, and Residual Networks, only ACNs exhibit auto-compression. Moreover, ACNs utilizing only the direct gradient (**DG**) still achieve significant auto-compression, highlighting the importance of a strong **DG** component to achieve this behavior[5] and explaining why FFNs do not exhibit auto-compression, as they lack a direct gradient term (Equation 4). In the case of Residual Networks, we previously argued that the exponential number of gradient paths substantially diminishes the influence of the direct gradient (**DG**)

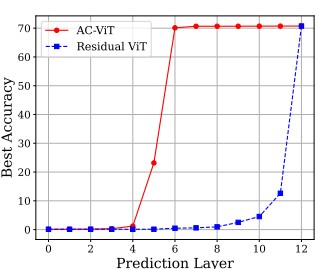

Figure 3: Performance of intermediate layers of AC vs Residual Vision Transformers trained on Imagenet-1K.

---

[5]ACNs with only the **DG** component under-perform, underpinning the importance of the **NG** component for maximizing performance.

on the overall gradient, a component crucial for auto-compression. To further illustrate this, Figure 2(right) presents the ratio of **DG** to the full gradient **FG** across layers during training for both AC and Residual variants. The results indicate a significantly higher **DG** to **FG** ratio in ACNs, confirming the increased contribution of direct gradients in the early layers of auto-compressing architectures compared to residual networks and explaining the auto-compression property. Furthermore, from Figure 1 we observe that ACNs demonstrate a concentrated gradient pattern with stronger signals in early layers and stronger patterns of **layer-wise learning**. Residual Networks exhibit a more "uniform layer learning" pattern, whereas deeper layers show increasing gradient contribution in later epochs, suggesting task-specific adaptation as training progresses. Interestingly, the pattern observed in Residual Networks indicates that high gradient norms are primarily concentrated in the early and deep layers, while middle layers receive significantly lower gradients, suggesting potential redundancy.

## 4    ACNs *compress* more

### 4.1    Auto-Compressing Vision Transformers and MLP-Mixers

Next, we evaluate ACNs in the context of transformer architectures by implementing an auto-compressing variant of Vision Transformer (ViT) [10]. We train a Vision Transformer (ViT) with long connections (AC-ViT) from scratch on the ILSRVC-2012 ImageNet-1K, following the training setup in the original paper. For both models we use 256 batch size due to memory constraints. AC-ViT converges at 700 epochs, while the Residual ViT converges at 300 epochs. As shown in Fig. 3, AC-ViT reaches top performance at only 6 layers while the vanilla ViT needs all 12 layers to reach similar performance, effectively suggesting that *ACNs can improve inference time and memory consumption without sacrificing performance*. To gain more intuition about the training dynamics and task learning of the two variants, in Figure 1(right) we plot the incremental layer performance contribution (difference in accuracy of subnetwork $i + 1$ to subnetwork $i$) to track the behavior of intermediate layers throughout training. The key observation is that ACNs are trained in a layer-wise fashion, while the residual variant performs task-learning in the last 2-3 layers, effectively utilizing the full network to achieve top performance.

**The effect of *Task Difficulty***    Intuitively, overparameterized networks trained on easier tasks should demonstrate higher levels of redundancy. Therefore, ACNs should converge to utilizing fewer layers as task difficulty decreases. To verify this, we use the number of classes as a proxy for task difficulty for image classification on the CIFAR-10 dataset [27]. Specifically, we create subsets of 2, 5, and 10 classes, the assumption being that binary classification should be easier than 10-class classification. For this experiment we utilize MLP-Mixer [50] and train two variants, the original MLP-Mixer with residual connections and the modified MLP-Mixer with long connections (AC-Mixer). Results are presented in Fig 4. We observe that indeed *AC-Mixer converges to solutions with larger effective depth, as the task "difficulty" increases*. Specifically, in this experiment, ACN needs 8, 10 and 12 layers for the 2, 5 and 10-class classification problem, respectively. In contrast, the Residual Mixer converges to solutions where the full depth of the network is utilized, irrespective of the task difficulty [6].

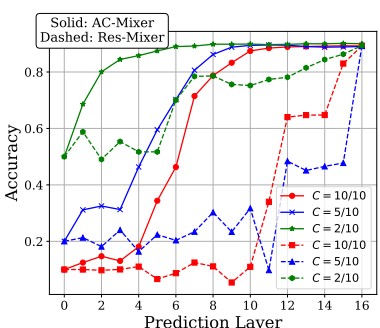

Figure 4: Performance of the intermediate layers as the number of classes (and examples) in the CIFAR-10 dataset increases from 2, to 5 to 10 classes: (a) Residual Mixer vs (b) AC-mixer (*C* denotes the number of classes in the subset).

### 4.2    Auto-Compressing Encoder Architectures for Language Modeling

In this section, we conduct a preliminary study on the effectiveness of the ACN architecture in general pre-training (masked language modeling with a BERT architecture) followed by fine-tuning. The

---

[6]The Residual Mixer was trained for 300 epochs, while AC-Mixer for 420 epochs to reach the performance of its residual counterpart.

results show that ACNs learn compact representations that: 1) achieve on-par performance with the residual architecture on transfer learning tasks, while utilizing significantly fewer parameters, and 2) complement post-training pruning techniques, enhancing their effectiveness.

### 4.2.1 Masked Language Modeling and Transfer Learning with ACNs

We compare the ACN and residual architectures in the standard BERT pre-training and fine-tuning paradigm. Using the original BERT pretraining corpus (BooksCorpus [64] and English Wikipedia), we train both architectures to equivalent loss values; the AC-BERT variant requires two epochs vs one epoch for the residual baseline. Following pre-training, we fine-tune both models on three GLUE benchmark datasets [54]: SST-2 sentiment analysis [45], QQP paraphrasing, and QNLI question answering [38]. Figure 5(left) demonstrates a key advantage of the ACN architecture: it naturally converges to using significantly fewer layers (approximately 75% less layers) while maintaining performance comparable to the full residual network. These results suggest promising applications for ACNs in large language models, where pre-training could be performed with long connections, allowing downstream tasks to adaptively utilize only the necessary subset of layers during fine-tuning. To further enhance compression, ACNs can be combined with standard pruning techniques—a preliminary investigation of this approach is provided in Appendix G.

## 5 ACNs *generalize* better

While ACNs demonstrate effective parameter reduction through architectural compression, a key question remains: do these compressed representations offer additional benefits beyond parameter efficiency? In this section, we investigate whether the concentrated information in ACNs' early layers leads to improved generalization capabilities compared to traditional residual architectures. Specifically, we explore robustness to input noise as a proxy for generalization. Moreover, we compare the inherent auto-compression of ACNs—achieved through architectural design—to recent regularization-based methods that rely on externally imposed intermediate losses.

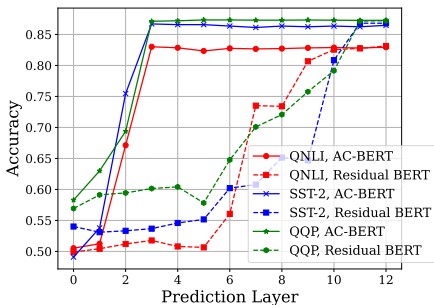

Figure 5: Downstream performance of AC-BERT vs residual BERT on three GLUE tasks.

### 5.1 Robustness to Input Noise

Next, we present results assessing the robustness of ACNs versus residual transformer architectures to input noise. The experiments are performed with the AC-ViT and residual ViT architectures trained on ImageNet-1K. In this experiment, we inject increasing levels of additive Gaussian noise with standard deviation $\sigma = 0.1, 0.2, 0.4$, and salt-and-pepper noise with percentage of altered pixels $p = 1\%, 2\%, 10\%$. Results (average accuracy) are shown in Table 2 (a) for Gaussian and (b) for salt-and-pepper noise. We observe that *ACNs display improved robustness to noise*, and the performance gap with the residual transformer increases as the noise levels increase. These results align with the findings of [58], who showed that architectures with forward passes closer to feedforward networks (like our ACNs) exhibit enhanced noise robustness. In residual architectures, short connections allow noise to propagate and accumulate throughout the network, whereas the long-connection design of ACNs helps mitigate this amplification effect.

| Model | Baseline | Gaussian Noise | | | Salt and Pepper Noise | | |
|---|---|---|---|---|---|---|---|
| | w/o noise | $\sigma = 0.1$ | $\sigma = 0.2$ | $\sigma = 0.4$ | $p = 0.01$ | $p = 0.05$ | $p = 0.1$ |
| Residual ViT | 70.74 | 67.68 | 62.80 | 45.46 | 56.80 | 27.48 | 10.34 |
| AC-ViT | 70.76 | 69.50 | 64.54 | 51.89 | 59.80 | 36.35 | 19.98 |

Table 2: Robustness (average accuracy %) of ViT with long connections (AC-ViT) and with residual connections (Residual ViT) to additive Gaussian noise and salt-and-pepper noise on ImageNet-1K test set.

## 5.2 Robustness to Data Sparsity

Next, we experimentally compare the performance of residual and long connections architectures in low-data scenarios. For this purpose, we create a random subset of CIFAR-10 [27] by retaining only 100 samples per class, resulting in a total of 1000 examples. Using the same training settings and models as described in Section 4.1 (MLP-Mixer on CIFAR-10), we train both architectures for 150 epochs to assess how fast the training and test loss decrease, as a proxy for the generalization capabilities of each architecture. Results shown in Fig. 6 reveal that ACNs achieve lower training and test loss in fewer epochs compared to residual networks. This faster convergence in loss metrics is a strong indication that auto-compressing networks can be effectively utilized in scenarios with limited data.

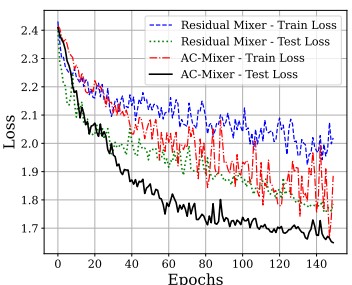

Figure 6: Train and Test Loss of AC-Mixer and Residual Mixer on CIFAR-10 (100 samples per class).

## 6 ACNs *forget* less

Continual learning involves training models on a sequence of tasks without access to past data, aiming to retain performance on previous tasks while learning new ones [8, 55]. A central challenge in CL is catastrophic forgetting—the tendency of neural networks to overwrite old knowledge when updated with new data. Common approaches include data replay methods [39, 33] and regularization techniques that penalize changes to important parameters [26, 61, 2]. We've already demonstrated that ACNs, through implicit layer-wise training, dynamically allocate parameters based on task demands while preserving redundant parameters for future tasks. Conversely, Residual Networks optimized for efficient task learning risk overfitting and suboptimal parameter usage in these sequential learning settings. To test our claims, we evaluate both architectures on the split CIFAR-100 continual learning benchmark, comprising 20 sequential disjoint 5-class classification tasks, focusing on task-incremental learning [51] where task identity is known. We utilize MLP-Mixer architectures (hyperparameters in Appendix B) and we test two continual learning algorithms trained for 10 epochs for each task: naive fine-tuning (Naive FT) and Synaptic Intelligence (SI) [61], which adds a gradient-based regularizer to each parameter depending on how changes in it affect the total loss in a task over the training trajectory. Across experiments, we report Average Forgetting, defined as the mean difference between a task's best performance (right after it is learned) and its final performance after all tasks are learned, and Average Accuracy, defined as the mean accuracy over all tasks at the end of training. We expect gradient-based regularization methods to perform particularly well with ACNs since unused, redundant parameters receive small gradients, making their detection easier compared to Residual Networks where gradients are more uniformly distributed (see gradient heatmaps, Fig. 1(left)). Results in Table 3 confirm our intuition: ACNs consistently exhibit significantly less forgetting (up to 18% improvement) compared to Residual Networks. Notably, with SI, increasing ACN depth decreases forgetting—an ideal behavior for CL systems where increasing network capacity reduces forgetting—while Residual Networks show the opposite pattern, indicating potential overfitting. ACNs also achieve better average accuracy across all tasks, further establishing them as a more suitable architecture for continual learning.

| Method | Arch | Avg. Accuracy (%) ↑ | | | Avg. Forgetting (%) ↓ | | |
| --- | --- | --- | --- | --- | --- | --- | --- |
| | | $L = 5$ | $L = 10$ | $L = 15$ | $L = 5$ | $L = 10$ | $L = 15$ |
| Naive FT | AC-Mixer | $32.97 \pm 2.4$ | $32.94 \pm 5.3$ | $31.61 \pm 2.2$ | $46.55 \pm 2.2$ | $45.46 \pm 5.8$ | $46.91 \pm 2.4$ |
| | ResMixer | $31.77 \pm 1.8$ | $28.16 \pm 1$ | $26.14 \pm 2.3$ | $52.76 \pm 2.3$ | $54.89 \pm 1.6$ | $54.49 \pm 2.2$ |
| SI | AC-Mixer | $44.5 \pm 2.2$ | $46.1 \pm 1.3$ | $\mathbf{46.2 \pm 0.8}$ | $35.7 \pm 2.1$ | $33.8 \pm 0.4$ | $\mathbf{32 \pm 1.8}$ |
| | ResMixer | $43.47 \pm 3.1$ | $36.1 \pm 5$ | $32.1 \pm 0.8$ | $42.4 \pm 4.1$ | $44.6 \pm 3.7$ | $50 \pm 2.1$ |

Table 3: Average accuracy and forgetting across layers, methods, and architectures on the Split CIFAR-100 continual learning benchmark. Models are trained for 10 epochs per task, where each task consists of classifying 5 out of 100 classes presented sequentially. $L$ denotes the number of layers in the architecture. ACNs consistently **forget less** and they also **do not waste capacity**.

# 7 ACNs *transfer* better

Parameter redundancy, and specifically potential layer redundancy, in residual architectures is a phenomenon that has been well documented [1, 53, 23]. Recent works [13, 24] have attempted to address this through regularization-based layer-wise compression techniques during training, specifically by adding losses to all intermediate layers of the network and using a weighted sum of them as the total loss, a technique formally introduced in [31] for improved training. Such loss-based regularization methods rely heavily on precise tuning of intermediate loss weights, creating practical challenges. If early-layer loss weights are set too high, the network risks overfitting and poor generalization; if set too low, performance improves gradually across layers with no clear cutoff point, reaching optimal results only at the final layer. This sensitivity to hyperparameter selection makes it difficult to reliably identify an optimal depth for inference using loss-based regularization. ACNs address this challenge through architectural design rather than regularization, naturally compressing information without requiring complex hyperparameter tuning.

To evaluate whether different layer compression approaches learn generalizable representations, we conduct a transfer learning experiment from CIFAR-100 to CIFAR-10 [27] using MLP-Mixer architectures (for hyperparameter choices we refer to Appendix B). This setup allows us to assess how well each model's learned representations transfer to a similar task. In regularization-based layer compression methods, explicitly training all layers to directly minimize a task loss through intermediate supervision can lead to overfitting on the base task, which can further re-

| Method | Accuracy (%) |
| --- | --- |
| AC-Mixer | **85.38 ±0.7** |
| Aligned | 82.9 ±0.9 |
| LayerSkip | 79 ±1.2 |

Table 4: C-100 to C-10 transfer learning performance of ACNs vs recent regularization-based layer copression approaches.

sult in weaker transfer capabilities on downstream tasks. In contrast, ACNs' implicit compression mechanism naturally balances generalizability and task performance without imposing external constraints. To ensure fair comparison, we train all models to achieve comparable performance on the CIFAR-100 pre-training task, enabling direct assessment of their transfer capabilities to CIFAR-10. The results in Table 4 confirm our analysis: ACNs demonstrate superior performance on the downstream CIFAR-10 task compared to regularization-based methods, even when upstream CIFAR-100 task performance is similar. This confirms our claim that the representations learned by ACNs are more generalizable and thus exhibit greater transferability.

# 8 Conclusion

In this work, we introduced Auto-Compressing Networks (ACNs), an architectural design that organically compresses information into early layers of a neural network during training via long skip connections from each layer to the output, a property we coined as *auto-compression*. Unlike residual networks, ACNs do not require explicit compression objectives or regularization; instead, they leverage architectural design and gradient-based optimization to induce implicit layer-wise training dynamics that drive auto-compression.

Our theoretical and empirical analyses demonstrate that ACNs alter gradient flow, imposing implicit layer-wise training dynamics and resulting in distinct representations compared to feedforward and residual architectures. In practice, this leads to 30–80% of upper layers becoming effectively redundant, enabling faster inference and reduced memory usage without sacrificing accuracy. Experiments across diverse modalities (vision, language) and architectures (ViTs, Mixers, BERT) further show that ACNs match or outperform residual baselines, while offering greater robustness to noise and low-data regimes, excelling in transfer learning, and reducing catastrophic forgetting by up to 18%—all without specialized tuning, overall suggesting that they learn better representations despite using fewer parameters. A summary of the main results is shown in Table 6. Another practical advantage of the auto-compression property of ACNs is its potential to enhance or complement other compression and efficient inference techniques. Preliminary results demonstrating this synergy are presented in Appendices G and H.

Concluding, ACNs highlight the potential of architectural design as implicit regularization and pave the way toward self-adapting neural networks that allocate depth and capacity dynamically. Future

work may extend ACNs to self-supervised, multi-task, and generative settings, as well as explore per-sample adaptive inference and other short vs long architectural variants. Some initial

## 9    Limitations and Broader Impact

Due to resource constraints, our evaluation of ACNs was limited to relatively small-scale tasks, though the architecture consistently performed well across modalities, datasets, and state-of-the-art baselines. Broader validation—including large-scale, self-supervised, and multi-task settings (e.g., language or multimodal models)—is essential to fully understand its capabilities and boundaries. A notable limitation is the increased training time compared to residual architectures. While this may contribute to stronger representations, optimizing training efficiency remains an open challenge, warranting further exploration of scheduling and initialization strategies.

By enabling implicit layer pruning, ACNs aim to reduce inference-time resource use, supporting more sustainable AI. Long-term, this could inspire adaptive architectures with System 1 / System 2 behavior [25], dynamically adjusting depth based on task complexity. However, like all efficiency advances, broader deployment may introduce ethical concerns—such as misuse in low-regulation environments or privacy-sensitive applications—highlighting the need for responsible development and oversight.

## 10    Acknowledgments and funding disclosure

This work has been partially supported by project MIS-5154714 of the National Recovery and Resilience Plan Greece 2.0 funded by the EU under the NextGenerationEU Program. We acknowledge EuroHPC JU project ID EHPC-AI-2024-A04-051 for use of the supercomputer LEONARDO@ CINECA, Italy.

G. Paraskevopoulos was funded by the European Union's Horizon Europe research and innovation programme under the AIXPERT project (Grant Agreement No. 101214389).

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

# Technical Appendices and Supplementary Material

## A   Extended Literature Review

We present here an extended literature review across four key areas relevant to our work: 1) residual connections and their role in training stability, 2) architectural variants with longer/denser residual connections, 3) methods that employ intermediate layer losses to learn better representations and exit early, and 4) neural architectures that induce regularization and better representations. Our work is more closely related to research area 4, but it is important to note that training stability, efficiency, performance, and representation learning are related goals, which can be achieved either through architectural choices (1, 2, 4) and / or through additional optimization criteria (3).

**Residual Connections and Training Stability:** Training deep neural networks with gradient descent becomes increasingly difficult as network depth increases. Multipath network architectures date back to the 1980s, with early work exploring cascade structures in fully connected networks trained layer by layer to improve training stability [15]. Highway networks [47] introduced gated bypass paths that allowed for effective training of networks with hundreds of layers. [20] found that deeper convolutional neural networks (CNNs) not only suffer from a decrease in generalization performance, often due to overfitting, but also experience a decrease in training performance. To address this, they introduced residual connections (or identity mappings), proposing that learning residual functions relative to identity mappings simplifies optimization. These skip connections improve the training process and often improve performance ([5], [36], [60], [32]). [53] further argued that a residual network with $n$ layers can be viewed as a collection of $2^n$ paths of varying length. At each layer, the signal either skips the layer or passes through it, creating $2^n$ possible paths. Despite sharing weights, these paths function as an ensemble of networks, as confirmed by experiments. In contrast, a traditional deep feedforward network has only one path, so removing any random layer significantly degrades performance. Additionally, the authors showed that these paths are typically shallow, with backward gradients often vanishing after passing through only a small fraction of the total layers.

**Residual Variants:** Following the success of residual networks various architectural modifications were proposed to improve efficiency and performance. In DenseNets [22], each layer is connected to all subsequent layers enabling for more efficient feature reuse; fusion is achieved through concatenation rather than addition. More recently, DenseFormer [37] introduced learned weighted averaging across layer outputs, while Depth-Wise Attention [14] applies attention mechanisms across block outputs.

**Intermediate Supervision and Early Exit:** In deeply supervised nets [31], complementary objectives are added to all intermediate layers to encourage hidden layers to learn more discriminative representations. In this approach, each intermediate objective $i$ is a loss function that captures the classification error of an SVM trained on the output features of layer $i$. The overall loss is the sum of the intermediate and final objectives. This idea evolved in several directions: Graves [17] proposed adaptive computation time, while more recent work like MSDNet [21] and CALM [44] introduced dedicated prediction heads. Other approaches employ trainable routing mechanisms [56, 57] to determine layer usage. Concurrent to our work, LayerSkip [12] proposes an architecture similar to ACNs, focusing primarily on inference acceleration through layer dropout and early exit mechanisms. Additionally, [24] also incorporates intermediate losses with a common head and a linearly increasing weight curriculum, justifying it through the lens of representational similarity between intermediate layers. While these approaches rely on explicit auxiliary objectives or dedicated components, our work achieves similar benefits through architectural design alone, enabling natural depth determination through gradient-based optimization.

**Architecturally-induced regularization and representation learning**: Stochastic regularization methods like Dropout [46] and its variants demonstrated that randomly dropping connections during training can lead to more robust feature learning. This insight was extended to other structural approaches like Stochastic Depth [23] where randomly dropping entire layers improved generalization. Residual connections initially proposed to address the vanishing gradient problem [20] have been shown to contribute to smoother loss landscapes and improved generalization [32]. These findings align with theoretical work showing that architectural choices impose implicit biases that influence the solutions found during training [19]. Recent work on transformers shows that architectural choices like attention patterns and layer normalization can also induce implicit regularization effects, e.g., the combination of skip connections and layer normalization can bias the model toward low-rank solutions

[4]. The proposed long connection approach builds on these insights, using architectural design to naturally encourage the learning of robust representation while enabling automatic information compression allowing for early exit.

## B  Experimental Details and Setup

**CIFAR-10 - MLP Mixer**: The MLP Mixers have 16 layers with a hidden size of 128. The patch size is 4 (the input is 32x32, 3 channels). The MLP dimension $D_C$ is 512, while $D_S$ is 64. We are using the AdamW optimizer [34] with a maximum learning rate of 0.001 and a Cosine Scheduler with Warmup. The batch size is 64.

**BERT post-training pruning**: For Magnitude pruning, we consider the setting where the pruning happens after fine-tuning on the downstream task. For Movement pruning, we follow a gradual fine-tune and prune curriculum, where in setting (I): 20% of the parameters are pruned after each epoch, whereas in setting (II): we prune 40% of the parameters after an epoch.

**Continual Learning Experiments**: We are using the same MLP-Mixer setup with the CIFAR-10 experiment (see above). We train for 10 epochs in each task, using AdamW with learning rate of 0.001 and a batch size of 64. For Synaptic Intelligence we use a coefficient $\lambda = 1$.

**Transfer Learning experiment Hyperparameter Choices:** In our main paper experiment we compare: 1) our proposed AC-Mixer, 2) a Residual Mixer with the setup of [24] (Aligned) and 3) a Residual Mixer with the setup of [13] (LayerSkip), with the rotational early exit curriculum with $p_{max} = 0.1$, $e_{scale} = 0.2$.

## C  Gradient Propagation equations derivation

**Notation**: $x_i$ is the output of layer $i$, $w_i$ is the weight of layer $i$ (the weight used to construct $x_i$), $x_0$ is the input (after a potential initial embedding operation) and $y_F$, $y_R$, $y_A$ is the output for each architecture.

FFN forward pass

$$y_F(= x_L) = \prod_{i=1}^{L} w_i x_0 \tag{2}$$

FFN backward pass for weight $i$

$$\frac{\partial y_F}{\partial w_i} = \frac{\partial y_F}{\partial x_i} \frac{\partial x_i}{\partial w_i} \tag{3}$$

$$\frac{\partial y_F}{\partial w_i} = \underbrace{\left( \prod_{k=i+1}^{L} w_k \right)}_{backward \text{ term}} \underbrace{\left( \prod_{m=1}^{i-1} w_m \right)}_{forward \text{ term}} x_0 \tag{4}$$

ResNet forward pass

$$x_i = w_i x_{i-1} + x_{i-1} = (1 + w_i) x_{i-1} \tag{5}$$

$$y_R = \prod_{i=1}^{L} (1 + w_i) x_0 \tag{6}$$

ResNet backward pass for weight $i$

$$\frac{\partial y_R}{\partial w_i} = \frac{\partial y_R}{\partial x_i} \frac{\partial x_i}{\partial w_i} = \left( \prod_{k=i+1}^{L} (1 + w_k) \right) \left( \prod_{m=1}^{i-1} (1 + w_m) \right) x_0 \tag{7}$$

$$\frac{\partial y_R}{\partial w_i} = \underbrace{\left(1 + \sum_{k=i+1}^{L} w_k + \sum_{i+1 \leq k < j \leq L} w_k w_j + \cdots + \prod_{k=i+1}^{L} w_k\right)}_{backward \text{ term}} \underbrace{\left(\prod_{m=1}^{i-1}(1 + w_m)\right) x_0}_{forward \text{ term}} \quad (8)$$

or equivalently:

$$\frac{\partial y_R}{\partial w_i} = \underbrace{\left(1 + \sum_{k=1}^{L-i+1} \texttt{sum of } \binom{L-i+1}{k} w \texttt{ k-tuples}\right)}_{backward \text{ term}} \underbrace{\left(\prod_{m=1}^{i-1}(1 + w_m)\right) x_0}_{forward \text{ term}} \quad (9)$$

ACN forward pass

$$y_A = x_0 + \sum_{i=1}^{L} x_i = x_0 + \sum_{i=1}^{L} \prod_{j=1}^{i} w_j x_0 \quad (10)$$

ACN backward pass for weight $i$

$$\frac{\partial y_A}{\partial w_i} = \frac{\partial y_A}{\partial x_i} \frac{\partial x_i}{\partial w_i} = \left(1 + \sum_{k=i+1}^{L} \frac{\partial x_k}{\partial x_i}\right) x_{i-1} \quad (11)$$

$$\frac{\partial y_A}{\partial w_i} = \underbrace{\left(1 + \sum_{j=i+1}^{L} \prod_{k=i+1}^{j} w_k\right)}_{backward \text{ term}} \underbrace{\left(\prod_{m=1}^{i-1} w_m\right) x_0}_{forward \text{ term}} \quad (12)$$

## D    Analysis of the Connectivity across architectures

Here, we further clarify the shown connections on Table 1. The red connections shown for ResNet connectivity, show the implicit direct connections (paths) that are formed in the architecture through the residual summation. Specifically, we denote:

- $z_0 = x_0$
- $z_{i-1}$: input to layer $i$ (for $i > 0$), which is multiplied with $w_i$ to get:
- $x_i = w_i \cdot z_{i-1}$: output of layer $i$ **before** residual summation
- $z_i = z_{i-1} + x_i$: output of layer $i$ **after** residual summation, or equivalently, input to layer $i + 1$

In our case, we want to express the input to the final layer $z_2$, which is:

$$z_0 = x_0$$
$$z_1 = x_1 + z_0 = x_1 + x_0$$
$$z_2 = x_2 + z_1 = x_2 + x_1 + x_0$$

Thus, we see that the residual summation $z_i = z_{i-1} + x_i$ effectively results in each layer $i$ receiving as input ($z_i$) a cumulative sum of all previous layer outputs ($x_*$). In standard feedforward networks (FFNs), each layer directly receives input only from the immediate previous layer. In contrast, in ACNs, intermediate layers behave like those in FFNs, but the final layer receives as input the outputs of all preceding layers.

# E Comparisons with Other Residual Architectures

To address concerns of overfitting and ensure fair comparisons in terms of training time, we hypothesize that ACNs tend to train longer due to a reduced number of information pathways compared to residual architectures (which scale linearly vs. exponentially in terms of connectivity). However, rather than overfitting, we argue that ACNs utilize this extended training period to learn more robust and generalizable representations.

To verify this, we conducted a controlled experiment using various architectures in the MLP-Mixer setting on CIFAR-10, training all models for 700 epochs to ensure fair comparison. The evaluated models include:

- **AC-Mixer:** An auto-compressing (ACN-style) Mixer,
- **Res-Mixer:** A vanilla residual Mixer,
- **DenseNet-Mixer:** A DenseNet-style variant [22], where each layer receives as input the concatenation of all previous layers' outputs, projected back to the hidden dimension via a learnable projection matrix. In order to have similar parameter count with the other architectures (since this architecture introduces additional parameters growing with depth), we reduce the number of layers to 10 (compared to 16 in the other models), and
- **DenseFormer-Mixer:** A DenseFormer-style variant [37], where each layer receives a learnable linear combination of all previous layers' outputs, with one scalar weight per previous layer ($O(L^2)$ extra parameters).

We report: (1) **Accuracy**, (2) **Best Acc. Epoch** — the epoch at which this peak accuracy was reached; and (3) **Cutoff Layer** — the earliest layer whose output reaches comparable performance to the final output, to measure auto-compression.

| Model | Accuracy | %Network used | Best Acc. Epoch |
|---|---|---|---|
| Res-Mixer | $90.3 \pm 0.09$ | 100% | 615 |
| DenseNet-Mixer | $90.3 \pm 0.10$ | 100% | 600 |
| DenseFormer-Mixer | $90.4 \pm 0.09$ | 100% | 637 |
| AC-Mixer | $\mathbf{92.0} \pm 0.08$ | 75% | 695 |

Table 5: Performance comparison across AC and residual variants on the MLP-Mixer/CIFAR-10 experiment. AC-Mixer converges slower but achieves the highest accuracy (improved generalization) with an earlier cutoff layer, showcasing stronger auto-compression.

The results in Table 5 show that AC-Mixer converges more slowly than residual variants but leverages the extended training to learn more compact and generalizable representations. Unlike residual and densely connected variants, which reach peak performance earlier, AC-Mixer achieves higher accuracy with fewer effective layers (earlier cutoff) and without signs of overfitting but rather stronger generalization, supporting our hypothesis.

# F Summary of Results

Table 6 presents the main results comparing residual (Res-) and auto-compressing (AC-) variants across both vision and language tasks. We observe that ACN variants consistently achieve comparable or superior accuracy while requiring significantly fewer parameters, lower computational cost at inference, and reduced storage requirements. These findings highlight the efficiency and scalability of the AC design, making it a strong alternative to traditional residual architectures.

# G Post-Training Pruning with AC-Encoders

ACN's primary advantage lies in its inherent compression capabilities during training, suggesting that when combined with pruning techniques, it should significantly outperform traditional residual architectures. To provide validation for this hypothesis, we conducted experiments using magnitude

| Model | Accuracy ↑ | #Params ↓ | GFLOPs ↓ | Size (MB) ↓ |
|---|---|---|---|---|
| Res-ViT on ImageNet | $70.74 \pm 0.09$ | 86M | 33.7 | 330 |
| AC-ViT on ImageNet | $70.76 \pm 0.12$ | 51M | 19.7 | 195 |
| Res-BERT on SST-2 | $86.63 \pm 0.09$ | 110M | 21.72 | 418 |
| AC-BERT on SST-2 | $86.68 \pm 0.06$ | 46M | 5.44 | 174 |
| Res-BERT on QNLI | $83.14 \pm 0.07$ | 110M | 21.72 | 418 |
| AC-BERT on QNLI | $83.07 \pm 0.10$ | 46M | 5.44 | 174 |
| Res-BERT on QQP | $87.20 \pm 0.09$ | 110M | 21.72 | 418 |
| AC-BERT on QQP | $87.30 \pm 0.07$ | 46M | 5.44 | 174 |

Table 6: Comparison of residual (Res-) and auto-compressing (AC-) variants across both vision and language tasks.

and movement pruning [42], two commonly employed baseline pruning techniques. Results are shown when fine-tuning of the SST-2 dataset sentiment analysis task. We refer to Appendix B for details regarding the experimental setup. Figure 7(right) confirms our hypothesis: ACNs consistently demonstrate superior compression-performance trade-offs compared to standard architectures, with their advantage becoming more pronounced at higher compression rates. This indicates that ACNs' architectural design naturally leads to more efficient parameter utilization, creating representations that are inherently more amenable to further pruning. While these preliminary results validate our approach to addressing parameter redundancy, they also point toward promising future directions. We anticipate that combining pre-trained ACN architectures with state-of-the-art pruning methods will result in extremely efficient, high-performing models, though rigorous validation of this hypothesis requires further investigation.

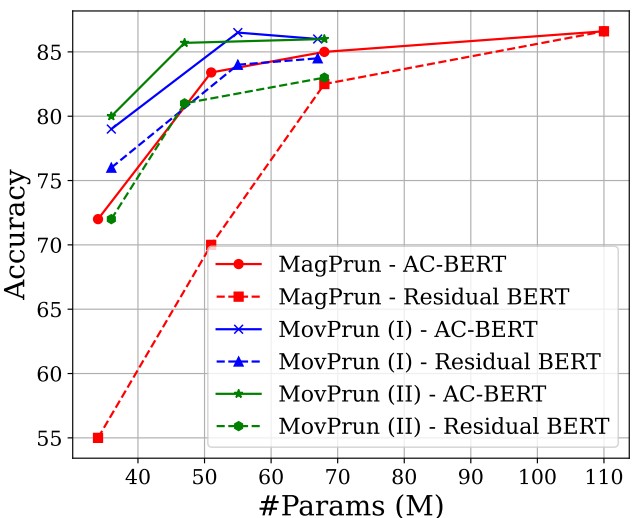

Figure 7: Accuracy vs model size of AC-BERT and Residual BERT on SST-2 when pruned with Magnitude and Movement Pruning (with two different settings, refer to Appendix B for details).

## H    Relation to Early Exit (EE) Methods

In this section, we explore the relationship between ACNs and Early Exit (EE) methods. While both aim to improve computational efficiency, ACNs are fundamentally different in their approach. Traditional EE methods are training-time strategies that rely on *explicit intermediate supervision*, whereas ACNs are architectural designs that achieve similar benefits *implicitly*, without auxiliary

losses or supervision. Importantly, these approaches are complementary: ACNs can be combined with EE techniques to further enhance both efficiency and flexibility.

**Overview of EE Methods.** Early Exit methods (e.g., BranchyNet [49]) typically:

1. Introduce explicit losses at intermediate layers to encourage early discriminative features.
2. Train using a weighted combination of intermediate and final losses.
3. Use confidence-based criteria (e.g., entropy) at inference to decide whether to exit early.

This line of work, originating from Deep Supervision [31], introduces several hyperparameters—such as loss weights—that require careful tuning. This tuning is crucial for the final behavior of the network since overemphasizing early-layer losses can lead to overfitting, as early layers are pushed to achieve low training loss.

In contrast, ACNs do not use any intermediate losses or auxiliary heads. They are trained end-to-end like standard feedforward or residual networks. Nevertheless, ACNs naturally produce more informative intermediate representations due to their implicit layer-wise compression dynamics. As demonstrated in Section 7, adding intermediate supervision to residual networks requires delicate balancing to avoid overfitting, while ACNs generalize more robustly—particularly in transfer settings—without any such tuning, thanks to their architectural regularization.

**ACNs complement EE methods.** Given that ACNs naturally concentrate discriminative information in earlier layers, they are well-suited for integration with EE mechanisms. To validate this, we conducted experiments on CIFAR-10 using MLP-Mixer variants, comparing six configurations:

- **Res-Mixer:** Standard residual MLP-Mixer,
- **AC-Mixer:** auto-compressing MLP-Mixer,
- **Res-Mixer w/ Branches:** Explicit early-exit branches at layers 4, 8, and 12, each with a dedicated MLP classifier,
- **AC-Mixer w/ Branches:** Same branching structure applied to the AC-Mixer,
- **Res-Mixer w/ Shared EE Head:** All layers connect to a shared classifier head, following recent works [24, 13], allowing exit decisions without additional parameters, and
- **AC-Mixer w/ Shared EE Head:** Same shared-head mechanism applied to the AC-Mixer.

For the **Branches** method, we use fixed weights $[0.8, 0.6, 0.4]$ for the three intermediate branches. In contrast, the **Shared EE Head** employs a linearly increasing layer weighting scheme, as in [24], defined by:

$$w_l = \frac{2(l+1)}{L(L+1)}, \quad \text{for } l = 0, 1, \dots, L-1.$$

During inference, we compute the entropy of the logits at each potential exit point. If the entropy falls below a pre-defined threshold ($\tau = 0.7$ for the results below), the model exits at that layer. Inference speedup is measured in terms of FLOPs, normalized to the baseline Res-Mixer.

| Model | Speedup (FLOPs) |
|---|---|
| Res-Mixer | $1.0\times$ |
| AC-Mixer (global cutoff at $L = 12$) | $1.5\times$ |
| Res-Mixer w/ Branches | $2.2\times$ |
| AC-Mixer w/ Branches | $2.6\times$ |
| Res-Mixer w/ Shared EE Head | $2.4\times$ |
| AC-Mixer w/ Shared EE Head | $\mathbf{3.3\times}$ |

Table 7: Inference speedup (measured in FLOPs) of various Early Exit configurations on CIFAR-10 using MLP-Mixer variants. All values are normalized to the FLOPs of the full Res-Mixer.

The results (Table 7) confirm that ACNs substantially improve the effectiveness of EE methods. When combined with a shared early-exit head, the AC-Mixer achieves the highest speedup—more than

$3\times$—benefiting from both implicit compression and confident early exits. This synergy highlights the strength of ACNs as a foundation for efficient inference.

We also draw a parallel between this behavior and the compatibility of ACNs with pruning methods (see Appendix G): in both cases, the compact representations of ACNs, through auto-compression, facilitate downstream efficiency techniques. Developing new early-exit strategies tailored to ACNs is a promising direction for future work.

