# OpenReview forum: "Auto-Compressing Networks"
_NeurIPS.cc/2025/Conference — NeurIPS 2025 oral_

### Official Review · Reviewer_ndrj · 2025-06-28

**Clarity:** 4
**Significance:** 4
**Originality:** 4
**Rating:** 5
**Confidence:** 4

**Summary:**

This paper proposes a new type of residual connection method, which makes the final network output directly combine all the previous layers' outputs. This network has the ability called "auto-compressing", which can identify the redundancy of layers in the neural networks.

**Questions:**

See weakness above.

**Ethical Concerns:**

["NO or VERY MINOR ethics concerns only"]

**Final Justification:**

This is a solid and interesting work, and the authors' responses have addressed most of my concerns. Thus, I maintain my accept score.

**Limitations:**

Yes

**Paper Formatting Concerns:**

No Concerns.

**Quality:**

4

**Strengths And Weaknesses:**

Strengths:
1. The paper writing is fluent and easy to read. The motivation is clear and easy to follow. The figures and tables are also clear.
2. The method is simple but effective, showing interesting "auto-compressing" properties of the network, which shows great potential in network compression.
3. The performance is good compared to the resnet counterpart.
4. The experiments are highly sufficient, showing their great potential in performance experiments, compression experiments, robust experiments, continual learning, and transfer learning.

Weaknesses:
1. Why Direct Gradient is so important in model training? It seems that its Direct Gradient makes the ACNs have the ability of the "auto-compressing". Do you have some hints of why?
2. It will be more convincing if you compare your method with some other variants of residual connection, like DenseNet.
3. Have you tried to further reduce the ACNs residual connections of some rear layers to further enhance the importance of the Direct Gradient to early layers and enhance the compression rate? Which I think can further verify your claims.

---

> ### Author Rebuttal · Authors · 2025-07-31
>
> ##  **Response to Reviewer**
>
> Thank you for taking time reading our manuscript and providing your insightful questions about the theoretical foundations of ACNs, particularly the role of Direct Gradient in driving auto-compression.
>
> ### **Key highlights of our response**
>
> - *Direct Gradient mechanism*: More detailed explanation of how direct gradient signals improve intermediate layer performance and drive information concentration
> - *Architectural comparisons*: Comprehensive comparison with DenseNet and other connectivity patterns as requested
> - *Gradient component analysis*: Experimental validation showing Direct Gradient drives auto-compression while Network-mediated Gradient remains important for performance
>
> ### **Detailed response**
>
> **The role of Direct Gradient:** Indeed , Direct Gradient (DG) is an important component for training, but it has to be  combined with the reduction of derivative flow through the residual paths for auto-compression to emerge (which is what happens in ACNs). We note also here that there is a line of work that investigates a similar idea (taking gradient signal directly from the output) as an alternative to backpropagation (see Direct Feedback Alignment).
>
> In our case, the significant contribution of the Direct Gradient (due to the linear number of paths) plays a crucial role in implicitly improving intermediate layer performance and driving the emergence of auto-compression. If we take as an example a classification task where $x_i$ is the output of layer $i$, the full output of the network is $y = x_L + \ldots + x_0$ (assuming $L$ layers), which is passed to a classification head to get a prediction. For a layer $i$, the information contained in the Direct Gradient dictates how this layer's weights need to change so that $x_i$, when summed to produce $y$, directly increases the performance. In this way, the partial sum up to layer $i$, $y_{\text{partial}} = x_i + \ldots + x_0$, is also directly improved (in terms of classification performance), effectively enhancing intermediate layer performance and driving information concentration in early layers. Another way to see this is by writing ACNs as $y = 1+w_1(1+w_2(1+w_3\ldots))$. This asymmetric form induced by ACNs, reveals that early layers are critical for the entire network---for example, a small $w_1$ makes all subsequent layers largely irrelevant. The DG signal facilitates training of these crucial early layers. In contrast, ResNets have the form $y = (1+w_1)(1+w_2)\ldots$, where the multiplicative structure with the +1 terms makes the network more robust to small weights in any individual layer.
>
> Thus, the Direct Gradient also facilitates early layers' learning at early stages of training, since the Network-mediated Gradient (NG) term consists of untrained weights and thus provides a noisy training signal, whereas the Direct Gradient, coming directly from the output, provides a more robust and reliable training signal early in training.
>
> **Regarding comparison with discussed related works:** We copy here for convenience the response to R1 (since word limit allows it), where we conducted an experiment including related works for comparison:
>
> We agree that comparison with discussed related works on different architectural ideas should be included. For this purpose, we conducted an experiment with the setup of MLP-Mixer on CIFAR-10 including:
>
> - an ACN style Mixer (AC-Mixer)
> - a Residual Mixer
> - a DenseNet(https://arxiv.org/abs/1608.06993) style Mixer, where each layer takes as input all previous layers' outputs, concatenates them, and projects them to *hidden_dim* with a learnable projection matrix per layer. For this variant, to keep the number of parameters the same as the others (because these projection matrices introduce extra parameters for each layer that also grow with depth), ensuring fair comparison, we set the number  of layers to 10 (instead of 16).
>  - DenseFormer(https://arxiv.org/abs/2402.02622) style Mixer, where each layer takes as input a linear combination of all previous layers' outputs with learnable scalars (each layer has one learnable scalar for each previous layer)
>
> To allow fair comparison, since ACNs are typically trained slower, we run all models for 700 epochs and report best accuracy (Acc.) and the (average across runs) epoch at which each model achieved it (Best Acc. Epoch). Furthermore, we report Cutoff Layer, which is the earliest layer that reaches top performance (compared to last):
> | Models | Accuracy | Cutoff Layer | Best Acc. Epoch |
> |--------|----------|--------------|-----------------|
> | Res-Mixer | 90.3±0.09 | 16 | 615 |
> | **AC-Mixer** | **92±0.08** | **12** | **695** |
> | DenseNet-Mixer | 90.3±0.1 | 16 | 600 |
> | DenseFormer-Mixer | 90.4±0.09 | 16 | 637 |
>
> From the results, we see that the AC-Mixer achieves the best performance (even better than the numbers reported in the main paper, due to increased training time), showcasing improved generalization despite the longer convergence time. Moreover, AC-Mixer is the only variant that exhibits Auto-Compression.
> We plan to include these results in the updated Appendix.
>
> **Reducing residual connections / Boosting Direct Gradient:** This is a valid and insightful point. In ACNs, each layer is connected directly to the next one and to the last one. The extra connections (non-direct), which are parts of the Full Gradient, are implicit and it is not straightforward to remove them. However, following the same line of thought, the general idea—which is intriguing—is that for a layer, one can essentially weight the backward signals, for example to enhance the Direct Gradient signal, effectively breaking the symmetry of backpropagation. While we did not test this approach, which would require careful implementation, we believe it is an interesting way to investigate the behavior of the different gradient signals.
> However, an experiment along the same lines is presented in Section 3, where we trained a variant of AC-Mixer that receives only the Direct Gradient as a backward signal (this setting represents the extreme case of what you are suggesting—removing all but the Direct Gradient signal). What we observed is that this variant exhibits strong Auto-Compression, highlighting the importance of that term in the emergence of this property. However, the model fails to match the performance of the other models, underscoring at the same time the importance of the NG component in task learning.
>
> We are happy to provide any additional analysis upon the reviewer’s request.

---

> > ### Comment · Reviewer_ndrj · 2025-08-01
> >
> > Thank you for your response. I have no further questions. This is an interesting and solid work. I’m inclined to maintain my original accept rating.

---

> > > ### Author Response · Authors · 2025-08-05
> > >
> > > We thank the reviewer for the kind feedback and for finding ACNs to be an interesting work. We appreciate your positive assessment and support for acceptance.

---

### Official Review · Reviewer_PWfu · 2025-07-02

**Clarity:** 3
**Significance:** 2
**Originality:** 1
**Rating:** 5
**Confidence:** 4

**Summary:**

This paper introduces Auto-Compressing Networks (ACNs), which replace traditional short residual connections with long feedforward connections from each layer directly to the output. The authors claim this architectural change leads to "auto-compression"—a property where networks organically concentrate information in early layers during training, making later layers redundant. They provide theoretical analysis of gradient dynamics and demonstrate that ACNs can achieve 30-80% layer reduction while maintaining performance across vision and language tasks, with some other benefits in noise robustness, continual learning, and transfer learning.

**Questions:**

1. can you give some head-to-head comparisons with MSDNet, BranchyNet, and EENet on identical datasets? Given the overlap, direct comparisons seem essential.
2. Can you show ACN results on full ImageNet and larger language models? The current small-scale experiments limit the generalisability claims.
3. From what I understand, training ACNs takes roughly twice as long as standard networks. When you factor in both the training overhead and inference costs, how do ACNs compare overall? I'm curious about the total computational budget, not just the inference savings - granted this is a one time cost.
4. There's already quite a bit of research on early exit mechanisms in neural networks. Beyond the specific way ACNs are architected, what makes them fundamentally different from existing early exit approaches? I'd like to understand the unique contribution here.

**Ethical Concerns:**

["NO or VERY MINOR ethics concerns only"]

**Final Justification:**

A thorough rebuttal demonstrating overall solid work. The paper makes a meaningful architectural contribution that complements existing efficiency methods.

**Limitations:**

Yes

**Quality:**

2

**Strengths And Weaknesses:**

Strengths

+ The gradient flow analysis (Table 1, Section 2.2) provides good intuition for why ACNs exhibit different training dynamics than ResNets, with the decomposition into Direct Gradient and Network-mediated Gradient components being interesting to read about.
+ The core auto-compression is demonstrated consistently across multiple architectures (ViT, MLP-Mixer, BERT) and datasets, showing the effect is not architecture-specific.
+ It's nice to see the authors also evaluate noise robustness, continual learning, and transfer learning, providing a broader picture of ACN behavior.
+ The architectural change is minimal and hardware-friendly, requiring no specialised software or complex hyperparameter tuning.

 Weaknesses

- The core contribution—networks with multiple prediction points that can exit early—has been extensively studied in the early exit literature (BranchyNet 2017, MSDNet 2017, EENet 2023). The paper fails to adequately compare with or even cite many key works in this space. MSDNet for example, already achieves "6% higher accuracy with 2-3× less computation time" on ImageNet
- Most experiments are on small-scale tasks (CIFAR-10/100, ImageNet with 256 batch size). For an architecture claiming broad applicability, this is insufficient. Early exit methods have been demonstrated at full ImageNet scale with superior results.
- ACNs seem to require significantly more training time (AC-ViT: 700 vs 300 epochs for ResNet). This negates claimed efficiency benefits and suggests the method may not scale to larger models where training cost matters more - unless I'm missing something?
- The paper claims "groundbreaking" results but often just matches baseline performance. Early exit methods like BranchyNet show 2-6× speedups, while ACN improvements are much more modest.
- No direct comparison to established early exit methods (MSDNet, BranchyNet, EENet) on the same tasks. This is a significant oversight given the clear conceptual overlap.
- Different convergence times between ACN and baseline variants make fair comparison difficult.
- Although it's great to see some transfer learning eval comparisons -they are pretty narrow with only CIFAR-100→CIFAR-10 transfer, which are very similar datasets.

---

> ### Author Rebuttal · Authors · 2025-07-31
>
> ##  **Response to Reviewer**
>
> We thank the reviewer for highlighting the early exit literature and pushing us to clarify ACNs' unique contribution. Your feedback helped us articulate that ACNs represent an architectural insight about connectivity patterns and  how these affect information flow and training dynamics, rather than another early exit method. We plan to include this discussion in the updated manuscript.
>
> ### **Our core argument**
>
> Our work reveals a fundamental architectural principle: direct-to-output connections coupled with reduced connectivity between layers (not "all-to-all" as in the case of residual networks) naturally compress information into early layers while maintaining performance—a property we term auto-compression. This insight transcends existing residual architecture variants by showing how connection topology intrinsically shapes representation learning. ACNs demonstrate that we can achieve ResNet-level performance with significantly fewer active layers and more robust representations (suitable for noisy and limited data tasks, as well as continual learning settings), opening new possibilities for hybrid architectures that adaptively combine short and long connections for optimal training dynamics, improved continual learning and deployment efficiency, particularly crucial for large language models.
>
> ### **Key highlights of our response**
>
> - *Architectural innovation vs. training strategy*: ACNs introduce a connectivity pattern that fundamentally changes gradient flow, complementing (not competing with) early exit (EE) training strategies
> - *Empirical validation of the insight*: ACN+EE outperforms ResNet+EE, proving that our connection pattern enhances existing methods
> - *Foundation for future work*: Our architecture opens new research directions in hybrid long/short connection patterns for large-scale models
> - *Training efficiency context*: Detailed analysis of training time trade-offs and their relationship to improved generalization
> - *Transfer learning validation*: Additional ImageNet→CIFAR-10 experiments confirming ACNs' representational advantages across domains
>
> ### **Detailed response**
>
> **Key Insight of our work:** The key insight of our work is not about early exit per se, but about how different connectivity patterns affect information flow, compression and ultimately representations during training - an insight that could help improve how large-scale architectures are designed.
>
> **Relation to Early Exit (EE) Methods:** ACNs are fundamentally different from Early Exit (EE) methods. While EE methods are training strategies that add explicit intermediate losses to existing architectures, ACNs are architectural backbones that achieve similar benefits through implicit design alone. Importantly, ACNs can be combined with EE methods to amplify their effectiveness.
> Typically  EE methods (e.g., see BranchyNet) methods follow a common pattern: (1) they add explicit intermediate losses at multiple layers to improve early feature discriminability, (2) these losses are weighted and combined with the final loss during training, and (3) confidence metrics like entropy are used at inference to decide where each sample should exit. This approach, pioneered by Deep Supervision (https://arxiv.org/abs/1409.5185), requires careful hyperparameter tuning of the loss weights.
>
> ACNs do not have any explicit regularizer in terms of extra intermediate losses or other regularizers, but instead they are trained "naturally," like residual or feedforward networks, and exhibit---through architectural design alone---improved intermediate layer performance (similar to intermediate losses) through implicit layer-wise dynamics.
>
> In Section 7, we show that coupling residual networks with these external intermediate losses requires careful tuning of the loss weights or it can result in potential overfitting (e.g., setting early losses to high weights can "overpush" early layers to achieve good predictions and results in overfitting on the training dataset). In practice, when compared with some recent works on intermediate losses, we showed that ACNs generalize better (as seen from improved transfer learning in the respective section) without requiring any extra hyperparameter tuning but instead relying on implicit architectural regularization.
>
> Nonetheless, ACN’s auto-compression property (pushing information naturally to earlier layers)  can be exploited by EE methods and magnify their effect. To empirically show this, we conducted an experiment on MLP-Mixer on CIFAR-10 where we compare:
> - a Residual Mixer
> - an AC-Mixer
>  -  a Residual Mixer with Branches (discussed below)
> - an AC-Mixer with Branches
> - a Residual Mixer with common EE head (discussed below)
> - an AC-Mixer with common EE head (discussed below)
>
> *Branches*: The idea is similar to BranchyNet. Specifically, we incorporate branches after layers 4, 8, and 12 with an MLP classifier to get intermediate branch losses that we use during training. After each of these layers, the outputs are passed to a dedicated classifier of the branch to get logits and calculate a loss that is combined with the common full network loss. At inference, we calculate the entropy of the logits of the branch classifiers and if it is below a certain threshold, we choose for this input to exit at the partlicular branch.
>
>  *Common EE head*: Based on more recent works [Aligned, LayerSkip - see paper], we get intermediate losses for each layer by passing its output to the common classification head and getting the logits. This has the advantage that no extra parameters are introduced, while fine-grained multiple exiting is achieved (since there is an exit at each layer---each layer is a branch). Again at inference, we calculate the entropy of the logits of an intermediate classifier and if it is below a threshold, then we exit at this layer.
> The results are shown below---the speedup is counted in terms of FLOPs relative to the FLOPs of the full Residual Network (16 layers):
>
> | Models | Speedup |
> |--------|---------|
> | Res-Mixer | 1x |
> | AC-Mixer | 1.5x |
> | Res-Mixer with Branches | 2.2x |
> | AC-Mixer with Branches | 2.6x |
> | Res-Mixer with shared EE head | 2.4x |
> | AC-Mixer with shared EE head | 3.3x |
>
> With 1$\times$ we have the Residual Network without EE exits, essentially utilizing the full network irrespective of the input. Then we see that ACN with Branches outperforms the residual variant with Branches, confirming our intuition that auto-compression can boost EE methods. Moreover, we see that coupling ACN with the common EE head method, which allows more fine-grained early exiting, provides a significantly larger speedup boost (and larger gap between ACNs and Residual Networks), showcasing the ability of ACNs to utilize different layers for different inputs and build strong EE schemes with  their internal classification head.
>
> Finally, we note that we view the behavior of ACNs with EE methods in a similar manner as ACNs coupled with pruning techniques that we showed in Appendix D: the inherent ability of ACNs to represent information better and more efficiently into fewer layers acts complementarily to pruning and EE  techniques and boosts their effect. Exploring new EE methods tailored for ACNs for even better efficiency is an exciting research direction that we are currently exploring.
>
> **Longer Training Times:**  We refer the reviewer to the related discussion with R1(**Discussion on Training time:**) and R2 (**Regarding potential overfitting and fair comparison in terms of training epochs**). Unfortunately, we cannot copy the responses here due to word limits.
>
> **Transfer Learning:** For testing the generalizability of the learned ACN representations we tested Cifar-100 to Cifar-10 transfer learning as you mentioned. We  also tested the BERT pre-training to fine-tuning (section 4.2) where we showed that ACNs achieve on-par performance with residual variants by utilizing 1/4th of the full network (3 vs 12 layers). Finally we present below results on transfer learning from imagenet (ViT) to Cifar-10:
>
> | Models | Acc.| Cutoff Layer|
> |--------|---------|---------|
> | Res-ViT | 94.09 |12 |
> | AC-ViT | 94.08 |6 |
>
> We see that again, ACNs achieve comparable performance with residuals while utilizing half of the network.
>
> **Scaling up:**
> While our current experiments are at moderate scale, the architectural insights about connection patterns become even more important for large models where both training and inference efficiency are critical. We are currently scaling these experiments to larger models and datasets, including full ImageNet training and large language models. Early results suggest that the auto-compression property becomes more pronounced at scale, making ACNs particularly promising for efficiency-critical large model deployment.
>
> We are happy to provide any additional analysis and discuss further upon the reviewer’s request.

---

> > ### Author Response · Authors · 2025-08-06
> >
> > We sincerely encourage the reviewer to consider our rebuttal and welcome any questions or comments you may have. Your feedback is appreciated, and we are glad to clarify any aspect that may require further explanation.

---

> ### Author Response · Authors · 2025-08-08
>
> We kindly invite the reviewer to engage in the discussion, review our rebuttal and share any additional questions or thoughts, especially as the discussion period is approaching its end. We would be happy to clarify any points from our rebuttal or provide further analysis where needed.

---

### Official Review · Reviewer_PUR5 · 2025-07-03

**Clarity:** 3
**Significance:** 3
**Originality:** 3
**Rating:** 5
**Confidence:** 3

**Summary:**

The proposed method, Auto-Compressing Networks, introduces a new architectural variant for neural networks where long feed-forward connections from each layer to the output replace shorter residual connections. This architectural modification is shown to organically compress information during training, with early layers capable of highlighting features that are sufficient for high performance, as well as number of other benefits like enhanced robustness to noise, superior performance in low-data settings, improved transfer learning capabilities and lower catastrophic forgetting.

**Questions:**

- Could the authors provide an in-depth comparison of the parameter count + FLOPs for the example of a single block (within any of the architectures used in the experiments) in addition to the overall training time required (for a chosen experiment)? A comprehensive comparison could help identify scenarios where the selection of the proposed architecture and its improved performance outweighs the cost.
- Could the authors provide notations on the graph of Table 1 that correspond to the notations used for forward and backward propagation?
This would help the reader quickly parse the table's information.
- Could the authors discuss the impact of hyper-parameter optimization on the experiments conducted in the paper (a subset would suffice)? The intention is to isolate the cause for improved performance and ensure the main reason is the change in architecture as opposed to sub-optimal hyper-parameter choice.
- I would recommend the authors provide the definition of "incremental layer performance" early on in the manuscript to ensure correct understanding of Fig. 1.
- Could the authors clarify how the gradient norm is normalized in Fig. 1?
- Could the authors double check MLM-Mixer in Fig. 2's caption? In addition, could the authors also double check the last 2 sentences in Section 3 (L. 171-176) and confirm whether the statements are correct?
- In Section 4.1, the manuscript mentions "ACNs can improve inference time and memory consumption without sacrificing performance". Are these exclusively measured using the layer vs. accuracy graphs? If so, from a practical standpoint, could the authors provide a params + FLOPs reduction count for the optimal layer setting as well? (Is this layer setting sufficient across all experiments?)
- Could the authors also discuss the potential of over-fitting when compared to Residual networks? (POV: Longer training time and smaller-scale datasets)
- Could the authors also provide a comparison where the residual architecture is trained for number of epochs = convergence epochs of ACN variant? In addition, could you comment on a potential causes + solutions to speed up convergence?
- Interestingly, in low-data settings, the convergence pattern is reversed. Could the authors discuss potential reasons for this?
- Due to the nature of the architectural variation, could the authors discuss and compare against early exit networks? (training strategy, stability, performance ?)
- To develop an even stronger appreciation for the changes affected by ACNs, could the authors discuss, using retrieval metrics or other similarity metrics, the feature representations learned by ACNs and Residual-variants?

**Ethical Concerns:**

["NO or VERY MINOR ethics concerns only"]

**Final Justification:**

The authors have provided comprehensive and detailed responses to all queries from my side in addition to questions from other reviewers as well. With the current state of responses in mind most of the remaining questions with relative short responses are more exploratory in nature.
With all of these in mind, my recommendation would be to accept the paper (with the caveat that all the manuscript level modifications are complete).

**Limitations:**

yes

**Quality:**

3

**Strengths And Weaknesses:**

Quality

The technical content and claims made in the paper are straightforward and well supported by experimental evidence. For individual comparisons, while the baselines are relevant they are not the current SoTA in all cases, slightly weakening the impact of the proposed model. Providing a more rounded discussion of the increase in size and training time and scenarios where ACNs might have drawbacks would serve to make this a very well rounded paper.

Clarity

The manuscript is extremely well written and clear, with sections dedicated to specific claims and supporting experimental results.

Significance

With the potential inclusion of FLOPs + Parameter counts and training time comparisons, the simplicity and implicit training dynamics  of ACNs could be extremely beneficial to the community at large, with the added bonus of offering larger compression and real-time deployment options.

Originality

The discussion about connection points with existing architectural variants and how ACN-specific changes affect training underline relevant hypotheses, novelty and how ACNs distinguish themselves.

---

> ### Author Rebuttal · Authors · 2025-07-31
>
> ##  **Response to Reviewer**
>
> Thank you for your constructive feedback on computational efficiency, practical deployment considerations, and the need for more comprehensive analysis. Your suggestions have helped us provide a more complete picture of ACNs' practical benefits.
>
> ### **Key highlights of our response**
>
> - *Comprehensive efficiency analysis*: Detailed FLOPs, parameter counts, and storage comparisons showing up to 4× reduction in inference costs
> - *Training vs. inference trade-offs*: Thorough analysis of roughly doubled training time against significant deployment benefits
> - *Fair comparison experiments*: All architectures trained for equal epochs (700) on CIFAR-10, confirming ACNs achieve better generalization (rather than overfitting) despite slower convergence
> - *Practical deployment focus*: Emphasis on one-time training cost vs. repeated inference savings for real-world applications
>
> ### **Detailed response**
>
> **Regarding FLOPs, Parameter counts and Training Time:** Below we present an analytical Table showing performance across runs, inference FLOPs, number of Parameters (after potential layer removal in ACNs) and Storage Size in MB for the ImageNet and BERT experiments:
>
> | Models | Accuracy ↑ | #Params ↓ | GFLOPs ↓ | Size in MB ↓ |
> |--------|------------|-----------|----------|--------------|
> | Res-ViT on ImageNet | 70.74±0.09 | 86M | 33.7 | 330 |
> | **AC-ViT on ImageNet** | **70.76±0.12** | **51M** | **19.7** | **195** |
> | | | | | |
> | Res-BERT on SST-2 | 86.63±0.09 | 110M | 21.72 | 418 |
> | **AC-BERT on SST-2** | **86.68±0.06** | **46M** | **5.44** | **174** |
> | Res-BERT on QNLI | 83.14±0.07 | 110M | 21.72 | 418 |
> | **AC-BERT on QNLI** | 83.07±0.1 | **46M** | **5.44** | **174** |
> | Res-BERT on QQP | 87.2±0.09 | 110M | 21.72 | 418 |
> | **AC-BERT on QQP** | **87.3±0.07** | **46M** | **5.44** | **174** |
>
> We see that consistently ACNs showcase significantly reduced GFLOPs (up to 4 times less) and reduced storage requirements without performance degradation, making them very well-suited for resource constrained real-world deployment.
>
> We will add this table to the updated Appendix.
>
> A block of an ACN is similar to a ResNet block in terms of computation, both having a residual summation:
>
> - ResNet block: y = f(x); x = y + x
>
> - ACN block: y = f(x); accum = accum + y
>
> In terms of training time, it's true that ACNs typically require more steps to converge, roughly double the steps in both the discussed experiments. We believe that this is a Limitation (as also mentioned in the related section of the paper and discussed in the response to R1) that requires further investigation. One interesting question is whether there exists an actual trade-off between training time and generalization capabilities (strength of learned representations), since it seems that ACNs utilize the increased training time to learn more robust representations often with improved generalization (see also experiment below). An interesting direction that we are currently exploring is combining the two architectures in order to improve training efficiency while keeping the auto-compression property that drives strong representation learning. All in all, we believe that since training is a one-time process, the significant inference benefits of ACNs along with the representational benefits (noise robustness, generalization) can potentially balance the increased cost of the training process.
>
> **Regarding potential overfitting and fair comparison in terms of training epochs:** Relevant to the previous discussion, we conducted an experiment with various architectures with the setup of MLP-Mixer on CIFAR-10, training all models for 700 epochs to ensure fair comparison (as requested also by R1). The involved models are:
> - an ACN style Mixer (AC-Mixer)
> - a Residual Mixer
> -  a DenseNet(https://arxiv.org/abs/1608.06993) style Mixer, where each layer takes as input all previous layers' outputs, concatenates them, and projects them to *hidden_dim* with a learnable projection matrix per layer. For this variant, to keep the number of parameters the same as the others (because these projection matrices introduce extra parameters for each layer that also grows with depth), ensuring fair comparison, we set the number  of layers to 10 (instead of 16).
>  - DenseFormer(https://arxiv.org/abs/2402.02622) style Mixer, where each layer takes as input a linear combination of all previous layers' outputs with learnable scalars (each layer has one learnable scalar for each previous layer)
>
> We report best accuracy (Acc.) and the epoch (averaged over runs) at which each model achieved it (Best Acc. Epoch). Furthermore, we report Cutoff Layer, which is the earliest layer that reaches top performance (compared to last):
> | Models | Accuracy | Cutoff Layer | Best Acc. Epoch |
> |--------|----------|--------------|-----------------|
> | Res-Mixer | 90.3±0.09 | 16 | 615 |
> | **AC-Mixer** | **92±0.08** | **12** | **695** |
> | DenseNet-Mixer | 90.3±0.1 | 16 | 600 |
> | DenseFormer-Mixer | 90.4±0.09 | 16 | 637 |
>
> The results show that  ACNs converge slower than all residual variants but utilize this increased training time to learn representations that generalize better and are more compact, rather than overfit to the dataset reaching better performance.
>
> **Regarding Early Exit methods** We provide a detailed  discussion of the relations between ACNs and the EE method in our response to R3; because of word limits in this response we cannot  copy the analysis here, we refer the reviewer to our response to R3 (Reviewer PWfu).
>
> **Comments on the Manuscript:**
>
> - For the notations of Table 1, we presented in more detail in the response of R1 what the dashed lines represent in the forward pass (direct connections). In the backward propagation, for ACNs and ResNets, for a layer i, the dashed lines (direct connections) denote the Direct Gradient path, where signals coming from subsequent layers form the Network-mediated gradient term. We presented the table this way for simplicity but we will consider your suggestions.
>
> - Thanks for the recommendation regarding the term "incremental layer performance" in Figure 1, we will add the definition to the caption of the Figure in the updated version. With  "incremental layer performance" of layer i, in ACNs we mean adding the output of layer i to the current sum (taking a partial sum from input embedding to this layer), passing it to the classifier (common head) and measuring performance. For ResNets, we get the output of layer i, add it to the residual stream, pass the result to the common classifier and get performance. So this heatmap essentially shows how the new computation induced by a layer increases the overall performance of the network (up to this layer).
>
> - In Figure 1 we take the L2 norm of the gradient of a layer. We will add this to the caption of the Figure.
>
> - In Figure 2 MLM-Mixer is a typo, the correct term is MLP-Mixer, thanks for noticing, we will update accordingly. Regarding (L. 171-176) there is an error in  the caption of  Figure 1  - we apologize for that. Specifically, the LEFT Figure (16 layers - 100epochs) is the MLP-Mixer on Cifar-10 experiment described in section 3 and the RIGHT figure is the AC-ViT on ImageNet experiment (12 layers - 300epochs). We will correct the caption and we apologize for the confusion. With this correction, in (L. 171-176) we note that as we see in Figure 1 (Left) the gradients of the residual network seem much more uniform compared to ACNs, with higher norms concentrating in the early and deeper layers.
>
> **Low Data settings**
> This was an interesting behavior that we observed. We hypothesize that this may be related to the distinct training dynamics of ACNs compared to ResNets. While ResNets are designed to utilize all parameters uniformly, ACNs may be more adaptive to data availability through their implicit layer-wise training dynamics. This is partly supported by our Task Difficulty experiment (Section 3), where ACNs appear to dynamically adjust capacity usage based on task complexity, while ResNets utilize all layers regardless. However, the exact mechanisms underlying this behavior are still under investigation.
>
> **Hyper-parameter optimization** We did not experiment extensively with hyperparameters, but rather used the default hyperparameters:   ViT on ImageNet from the original paper, and BERT pretraining and fine-tuning).  While experimenting further with hyperparameters  might improve ACN performance or convergence speed, we show that using off-the-self vanilla resnet hyperparameter values provide good results. One observation worth noting is that in some cases, ACNs benefit from higher warmup and/or lower learning rates compared to Residual Networks. This more conservative setup likely helps facilitate the implicit layer-wise dynamics found in ACNs.
>
> **Representations in ACNs vs ResNets:** We have an active research thread on mechanistic interpretability for ACNs and hope to be able to report to the community soon our findings. We plan to include a more detailed analysis of feature representations using similarity metrics and retrieval methods in our revised manuscript. Preliminary analysis suggests ACNs learn more compressed yet discriminative representations in early layers.
>
> We are happy to provide any additional analysis and discuss further upon the reviewer’s request.

---

> > ### Author Response · Authors · 2025-08-06
> >
> > We respectfully invite the reviewer to examine our rebuttal and share any additional thoughts, questions, or concerns. We truly value your insights and are willing to clarify any points that may need further elaboration.

---

> ### Comment · Reviewer_PUR5 · 2025-08-07
> **Follow Up**
>
> I am extremely happy to note your detailed responses to the questions in the review. Below, I highlight a few minor points that could be useful to keep in mind.
> - Practical deployment cost: While I agree that the cost of training is only borne once, when adapting to different domains this becomes critical. In addition, following threads that could potentially accelerate the learning of ACNs could prove to be extremely beneficial. (HPO)
> - Low data availability: Following up on the low data setting, could the authors connect the idea of how ACNs build robustness through the latter part of their training, the adaptive capacity usage and its impact on the low data setting. Is there an empirical way to verify the hypothesis?
> - Representation in ACNs vs. ResNets: As a minor follow-up, could the authors clarify how they measure the level of compression in representations?
>
> Overall, with the responses I am happy to increase my score.

---

> > ### Author Response · Authors · 2025-08-07
> >
> > We thank the reviewer for the insightful feedback, which helped us improve the manuscript and better highlight the unique characteristics of ACNs compared to other architectures. We also appreciate your consideration in increasing the initial score.
> >
> > ## Regarding your follow-up questions:
> >
> > - **HPO**: We agree that training time is an important consideration, even if it's a one-time cost. We also believe that more tailored hyperparameter optimization for ACNs can help reduce the training time gap—a direction we are actively exploring.
> >
> > - **Low data / Robustness**: We believe both behaviors arise from the structure of the networks, particularly the number of residual paths (linear in ACNs vs exponential in ResNets). This affects:
> >
> >  **a)** Robustness: In ResNets, residual connections pass input noise through all layers. This is supported by [1], which shows that interpolating between residual and feedforward networks improves noise robustness (we note that ACNs have forward pass similar to FFNs and backward closer to ResNets).
> >
> >  **b)** Low-data performance: The structured path design in ACNs induces implicit layer-wise training dynamics, allowing layers to adapt based on task needs—potentially explaining better performance with sparse data. This could be tested by training smaller ResNets (matching ACNs' converged depth) or applying explicit layer-wise training to ResNets/FFNs.
> >
> > - **Representations:** As a proxy for compression, we use the number of active layers. For instance, if an ACN uses only 6 of 12 layers to match the performance of a 12-layer ResNet, we consider this indicative of more compact representations (fewer parameters). Investigating compression via information-theoretic metrics is an interesting direction for future work.
> >
> > [1] https://arxiv.org/abs/2006.05749

---

> > > ### Comment · Reviewer_PUR5 · 2025-08-07
> > > **Follow up**
> > >
> > > Thank you for your responses. I do not have any further questions at the moment.

---

### Official Review · Reviewer_DqLE · 2025-07-06

**Clarity:** 3
**Significance:** 3
**Originality:** 3
**Rating:** 5
**Confidence:** 4

**Summary:**

This work proposes a new network architecture that includes long skip connections from each layer to the output. The authors demonstrate that this design outperforms traditional skip-connection networks across diverse modalities and training settings, including supervised learning, low-data regimes, and task-incremental learning.

**Questions:**

* Why do ACNs show better generalization? Are there any theoretical explanations or underlying intuitions that support this behavior?
* How many runs were performed for each experiment? Please report mean and variance if applicable.
* Line 183: "AC-ViT converges at 700 epochs, while the Residual ViT converges at 300 epochs." Does this imply that Residual ViT converges faster than AC-ViT? If so, can you clarify the implications for training efficiency? Please include learning curves if possible. Additionally, do all residual networks converge faster than ACNs across tasks?

**Ethical Concerns:**

["NO or VERY MINOR ethics concerns only"]

**Final Justification:**

The rebuttal addressed all my concerns. Thus, I incline to accept this work.

**Limitations:**

Yes.

**Paper Formatting Concerns:**

None.

**Quality:**

3

**Strengths And Weaknesses:**

### Strengths

* Well-designed experiments are provided, offering detailed analysis that directly supports the authors' claims.
* Experimentally, the proposed method outperforms residual networks in reducing forgetting, improving generalization, increasing robustness to input noise, and achieving better capacity utilization.

### Weaknesses

* The equations and figures in Table 1 appear to be incorrect. For example, in the ResNet architecture, there should not be a connection between the first and last layers. Additionally, activation functions are missing from the equations. In the case of FFN, the final output $y_F$ cannot be obtained as currently formulated unless only linear activations are assumed. This omission likely affects all derivations in Appendix C.
* In most experiments, the proposed method is compared only against ResNet. Ideally, additional baselines—such as those mentioned in the related work section—should be included for a more comprehensive evaluation.
* Table 4 lacks error bars, which limits the interpretability and reliability of the reported results.

I am open to increasing the score once these limitations are addressed. If corrected, this paper has the potential to make a meaningful contribution to the community.

---

> ### Author Rebuttal · Authors · 2025-07-31
>
> ##  **Response to Reviewer**
>
> We thank the reviewer for the detailed technical feedback on our mathematical formulations, experimental setup, and theoretical foundations. Your questions helped us clarify key aspects of ACNs and strengthen our experimental validation.
>
> ### **Key highlights of our response**
> - *Mathematical clarification*: We've clarified that the equations in Table 1 refer to an 1D linear network, while the connectivity diagram shows the 2D linear case.
> - *Expanded baselines*: New comprehensive comparison including DenseNet and DenseFormer variants on CIFAR-10, showing ACNs achieve the best performance with auto-compression.
> - *Error bars added*: Updated Table 4 with proper statistical reporting (mean ± variance).
> - *Theoretical insights*: Detailed explanation of why ACNs show better generalization through more effective depth utilization and robust layer-wise training dynamics.
>
> ### **Detailed response**
>
> **Regarding Table 1:**  The mathematical analysis for clarity and simplicity is conducted for 1D linear neural networks of depth L as mentioned in line 98 and in the caption of Table 1. However, the connectivity is shown for a 2D linear case. We apologize if this was not clear for the connectivity and we will modify the caption of Table 1 to state it explicitly (2D case -> 2D linear case for connectivity diagram vs 1D linear case for the equations).
>
> To clarify a bit further: with the red connections in the case of ResNet connectivity we show the implicit direct connections (paths) that are formed in the architecture through the residual summation.  We denote:
> - $z_0 = x_0$
> - $z_{i-1}$: input to layer $i$ (  for $i>0$), that is multiplied with $w_i$ to get:
> - $x_i = w_i \cdot z_{i-1}$: output of layer $i$ (BEFORE residual sum)
> - $z_{i} = z_{i-1} + x_i$: output of layer $i$ AFTER residual sum, or equivalently input to layer $i+1$
>
>
> So in our case, we want to express the input to the final layer $z_2$, which is:
>
> - $z_0 = x_0$
> - $z_1 = x_1 + z_0 = x_1 + x_0$
> - $z_2 = x_2 + z_1 = x_2 + x_1 + x_0$
>
>
> We will add this detailed explanation in the updated Appendix.
>
> **Regarding comparison with discussed related works:** We agree that comparison with discussed related works on different architectural ideas should be included. For this purpose, we conducted an experiment with the setup of MLP-Mixer on CIFAR-10 including:
> - an ACN style Mixer (AC-Mixer)
> - a Residual Mixer
> - a DenseNet(https://arxiv.org/abs/1608.06993) style Mixer, where each layer takes as input all previous layers' outputs, concatenates them, and projects them to *hidden_dim* with a learnable projection matrix per layer. For this variant, to keep the number of parameters the same as the others (because these projection matrices introduce extra parameters for each layer that also grow with depth), ensuring fair comparison, we set the number  of layers to 10 (instead of 16).
>  - a DenseFormer(https://arxiv.org/abs/2402.02622) style Mixer, where each layer takes as input a linear combination of all previous layers' outputs with learnable scalars (each layer has one learnable scalar for each previous layer)
>
> To allow fair comparison, since ACNs are typically trained slower, we run all models for 700 epochs and report best accuracy (Acc.) and the (average across runs) epoch at which each model achieved the best accuracy (Best Acc. Epoch). Furthermore, we report Cutoff Layer, which is the earliest layer that reaches top performance (showcasing auto-compression):
>
> | Models | Accuracy | Cutoff Layer | Best Acc. Epoch |
> |--------|----------|--------------|-----------------|
> | Res-Mixer | 90.3±0.09 | 16 | 615 |
> | **AC-Mixer** | **92±0.08** | **12** | **695**|
> | DenseNet-Mixer | 90.3±0.1 | 16 | 600 |
> | DenseFormer-Mixer | 90.4±0.09 | 16 | 637 |
>
> From the results, we see that the AC-Mixer achieves the best performance (even better than the numbers reported in the main paper, due to increased training time), showcasing improved generalization despite the longer convergence time. Moreover, AC-Mixer is the only variant that exhibits Auto-Compression.
> We plan to include these results in the updated Appendix.
>
> **Regarding Table 4:** We present below the updated table with the error bars and we plan to also update the Table in the updated version of the paper.
>
> | Models | Acc.|
> |--------|---------|
> | **AC-Mixer** | **85.38±0.7** |
> |Aligned | 82.9±0.9|
> | LayerSkip | 79±1.2 |
>
> **ACN vs ResNet generalization capabilities:** While Residual Networks were introduced to offer efficient and effective training especially in very deep architectures, there are various works that explore behaviors of these architectures that may harm generalization:
> - In [1, 2] it is shown that these architectures exhibit a notable resilience to layer dropping and permutation, which could indicate potential redundancy of some layers (i.e. removing a layer does not affect the network).
> - In [3], it was further observed that dropping subsets of layers during training can reduce overfitting and improve generalization.
> - In a related study [4], the authors showed that introducing skip connections between layers can lead to parts of the network being effectively bypassed and under-trained.
> - In [5], they show that unscaled residual connections can harm generalization capabilities in generative representation learning.
> - More recently, research has revealed substantial parameter redundancy and inefficient parameter usage in large-scale foundation models, particularly within their deeper layers [6] and  also [7].
>
> All these observations can be unified under the perspective that, although residual architectures facilitate training via multiple signal pathways, these same pathways
> can sometimes act as shortcuts that cause certain components to be either underutilized or prone to overfitting—ultimately limiting effective generalization. That means that vanilla residual architectures may not utilize their depth that effectively. ACNs have the potential of achieving better representations through direct derivative flow to each layer and the removal of skip connections not allowing layers to be bypassed during training. In this work, we experimentally show that ACNs  can match Residual Networks task performance while utilizing significantly fewer layers, while the representations learned can exhibit extra benefits such as noise robustness.  A detailed analysis of mechanistic interpretability of each architecture is left as future work.
>
> [1] https://arxiv.org/abs/1605.06431, [2] https://arxiv.org/abs/2407.09298, [3] https://arxiv.org/abs/1603.09382, [4] https://arxiv.org/abs/1610.01644, [5] https://arxiv.org/abs/2404.10947, [6] https://arxiv.org/abs/2403.17887, [7] https://arxiv.org/abs/2505.13898
>
> **Detailed summary of results:** We present in the table below mean and variance for the main experiments (along with parameter counts, flops, storage size)
>
> | Models | Accuracy ↑ | #Params ↓ | GFLOPs ↓ | Size in MB ↓ |
> |--------|------------|-----------|----------|--------------|
> | Res-ViT on ImageNet | 70.74±0.09 | 86M | 33.7 | 330 |
> | **AC-ViT on ImageNet** | **70.76±0.12** | **51M** | **19.7** | **195** |
> | | | | | |
> | Res-BERT on SST-2 | 86.63±0.09 | 110M | 21.72 | 418 |
> | **AC-BERT on SST-2** | **86.68±0.06** | **46M** | **5.44** | **174** |
> | Res-BERT on QNLI | 83.14±0.07 | 110M | 21.72 | 418 |
> | **AC-BERT on QNLI** | 83.07±0.1 | **46M** | **5.44** | **174** |
> | Res-BERT on QQP | 87.2±0.09 | 110M | 21.72 | 418 |
> | **AC-BERT on QQP** | **87.3±0.07** | **46M** | **5.44** | **174** |
>
> We will add this table to the updated Appendix.
>
> **Discussion on Training time:** We found that in most cases ACNs do converge slower than residual networks (sometimes needing double the number of steps). We believe that this might be due to the direct derivative flow to the attention modules making inter-layer coordination harder to achieve, as well as a smaller number of information pathways that facilitate information flow. We view the increased training time as a limitation of the architecture that is outweighed by better and more compact representations.   An active research direction that we are exploring is how to combine both architectures to achieve improved training efficiency while enjoying the benefits of Auto-Compression. Moreover, since training represents a one-time upfront cost while inference occurs repeatedly throughout a model's deployment lifecycle, the substantial inference speedups make this trade-off favorable for real-world applications.
>
> We are happy to provide any additional analysis upon the reviewer’s request.

---

> > ### Comment · Reviewer_DqLE · 2025-08-04
> >
> > Thank you for your detailed response. I'm happy to increase my score.

---

> > > ### Author Response · Authors · 2025-08-05
> > >
> > > We thank the reviewer for the thoughtful feedback and for considering increasing the initial score.

---

### Decision · Program_Chairs · 2025-09-17

**Decision:**

Accept (oral)

**Comment:**

This paper introduces auto-compressing networks, with feedforward connections from each layer to the final layer. These networks are capable of generalizing better than typical residual networks, and have enhanced transfer learning capabilities. The reviews for this paper are excellent. Any concerns reviewers expressed were thoroughly addressed through the rebuttal process. A new method that out-performs residual networks will have wide ranging impacts for the community.